# LieGG: Studying Learned Lie Group Generators

**Artem Moskalev**
UvA-Bosch Delta Lab
University of Amsterdam
a.moskalev@uva.nl

**Anna Sepliarskaia**
Machine Learning Research Unit
TU Wien
seplanna@gmail.com

**Ivan Sosnovik**
UvA-Bosch Delta Lab
University of Amsterdam
i.sosnovik@uva.nl

**Arnold Smeulders**
UvA-Bosch Delta Lab
University of Amsterdam
a.w.m.smeulders@uva.nl

## Abstract

Symmetries built into a neural network have appeared to be very beneficial for a wide range of tasks as it saves the data to learn them. We depart from the position that when symmetries are not built into a model a priori, it is advantageous for robust networks to learn symmetries directly from the data to fit a task function. In this paper, we present a method to extract symmetries learned by a neural network and to evaluate the degree to which a network is invariant to them. With our method, we are able to explicitly retrieve learned invariances in a form of the generators of corresponding Lie-groups without prior knowledge of symmetries in the data. We use the proposed method to study how symmetrical properties depend on a neural network's parameterization and configuration. We found that the ability of a network to learn symmetries generalizes over a range of architectures. However, the quality of learned symmetries depends on the depth and the number of parameters.

## 1 Introduction

Convolutional Neural Networks (CNNs) are efficient, for one because they can convert the translation symmetry in the data into a built-in translation-equivariance property of the network without exhausting the data to learn the equivariance. Group-equivariant networks generalize this property to rotation [6, 42, 39, 20], scale [35, 33, 4], and other symmetries defined by matrix groups [14]. Equipping a neural network with prior known symmetries has proved to be data-efficient.

Recent works [13, 21, 46] have demonstrated that hard coding the symmetry into a neural network does not always lead to a better generalization. Often, soft or learned equivariance is more advantageous mostly in terms of data efficiency and accuracy [38]. Moreover, architectures with no equivariance built-in, such as transformers [11] and multi-layer perceptron mixers [36], achieve a remarkable performance on a wide range of problems. Effectively, they learn their own geometrical priors without explicit symmetry constraints [8]. This raises the following questions to study in this paper. To what degree do neural networks learn symmetries? How accurately do learned symmetries reflect the true symmetries in the data? Can we support the capability of neural networks for learning symmetries? We present a method to study symmetries learned by neural networks to address the questions.

In the paper, we depart from Lie group theory and develop a method that can retrieve symmetries learned from the data by any neural network (Figure 1). Our method only makes the assumption that

Source code: https://github.com/amoskalev/liegg

36th Conference on Neural Information Processing Systems (NeurIPS 2022).

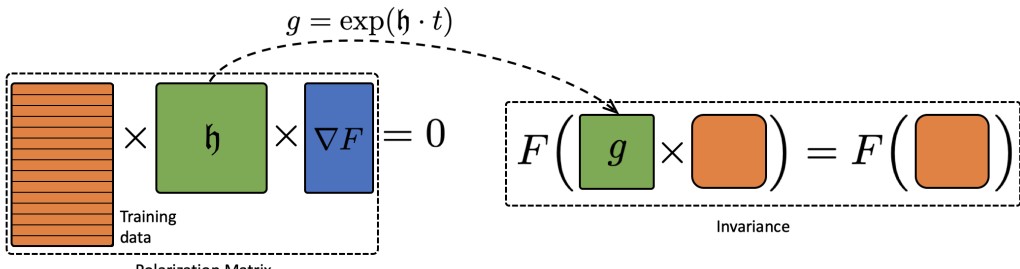

Figure 1: We solve the matrix nullspace equation to derive an infinitesimal generator from a neural network and the training data. Due to Lie algebra - Lie group correspondence, we can calculate the class of transformations the model is invariant to by exponentiating the generator.

a model is differentiable, commonly met in neural networks. While previous works on analyzing symmetries in neural networks rely on empirical analysis of network representations [28] or on examination of a given set of transformations [15, 33, 25], our method outputs a generator of the corresponding Lie-group and allows to quantitatively evaluate how sensitive the network is in the direction of the learned symmetry We make the following contributions:

- From the perspective of Lie-groups, we propose the theory to study symmetrical properties of neural networks.
- We derive an efficient implementation of the method that allows improving the interpretability of the model by revealing symmetries it learned.
- With our method we conclude that models with more parameters and gradual fine-tuning learn more precise symmetry groups with a higher degree of invariance to them.

## 2 Related work

**Equivariant and invariant networks**    The goal of equivariant and invariant networks is to build the symmetries into a neural network architecture as an inductive bias. Starting from the convolutional networks [24] that contain a translation symmetry, the concept of the equivariance was generalized to rotations [6, 42, 39, 20], permutations [44], scaling [35, 33, 34, 4, 41] and arbitrary matrix groups in [14]. Various methods have been proposed for learning symmetries directly from the training data [46, 3] among which a popular approach is to learn transformations by estimating the infinitesimal generators of symmetry groups [32, 10, 7, 31, 9]. These papers focus on building the symmetries into the models by modifying the architecture. In our work, we focus on the reverse question, i.e. given a network with a fixed architecture, what symmetries does it learn from the data?

**Symmetries in neural networks**    Another line of work is focused on interpreting symmetries in neural networks. In [28] Olah *et al.* take inspiration from biological circuit motifs [2] and study equivariant patterns learned by an unconstrained neural network on the image classification task. The authors demonstrate that the network learns rotation, hue, and scale symmetries when trained on ImageNet. In [15] Goodfellow *et al.* propose a number of empirical tests to study invariances to known transformations in a network, and demonstrate that auto-encoding architectures learn increasingly invariant features in the deeper layers. In [25] Lenc & Vedaldi quantify invariances and equivariance in the layers of convolutional networks to a pre-defined set of transformations. In the concurrent work of Gruver *et al.* [16], the authors propose using the Lie derivative to measure the local equivariance error to known symmetry groups. In contrast to these works, our method does not require knowing the set of transformations beforehand; and it also provides theoretical guarantees of the invariance from the perspective of the Lie groups theory.

**Network configuration: width, depth, number of parameters**    Neural networks of various widths and depths have been studied through the lens of universal approximation theorem [26, 22], functional expressiveness [30] and by empirical analysis of learned representations [27]. These works focus on either characterizing learning capabilities of neural networks or on interpreting the differences

between representations that models with various architectures learn. They, however, do not analyze how the symmetrical properties of a model depend on its width and depth.

Other works focus on analyzing a connection between a number of parameters and the generalization capability of networks [1, 43, 37, 45]. It is commonly argued that overparametrized architectures achieve better generalization bounds, i.e. provide smaller discrepancy between train and test performance and are thus preferred in many applications [45]. However, it is unclear if the models with more parameters and hence with a finer generalization capability also learn symmetries better. In our paper, we investigate this question for a family of feed-forward networks.

**Robust Learning**    Various methods have been proposed to train more robust neural networks. While a naive training may lead to overfitting on the training subset, a well-organized pre-training helps to mitigate this issue [12]. Features extracted by a model pretrained on a bigger dataset often demonstrate more robust results [18]. While there are many effective techniques for training more robust feature extractors [5, 47, 17], the analysis of such methods from the invariance point of view has not been done before. We demonstrate that our method allows for better understanding and explaining what training regimes lead to more invariant, and thus more robust representations.

# 3   Background

The focus of this paper is a *symmetry group*. A symmetry is a transformation of an object that leaves the object unchanged. The set of all such transformations of an object with composition as a binary operation forms a group. In this paper, the object of interest is a dataset and its symmetries. We study what kind of transformations map one data sample to another and how neural networks learn information about the symmetries.

The theory of Lie groups is a sweet spot of mathematics that helps to formalize symmetries and provides practical methods for studying them. Formally, a Lie group is a group that has the structure of a differential manifold. The tangent space in the identity element of a Lie group forms a vector space called Lie algebra. A Lie algebra determines a Lie group up to an isomorphism for simply connected Lie groups and, being a vector space, is a more convenient object to study than a Lie group. Thus, in order to recover a simply connected symmetry group, it is sufficient to understand its Lie algebra.

In this paper, we assume that the Lie group $G$ is a matrix Lie group, i.e. is a closed subgroup of the general linear group of the degree $n$, denoted as $GL(n)$. It is defined as the set of $n \times n$ invertible matrices. The correspondence between the Lie group $G$ and its Lie algebra $\mathfrak{g}$ in this case is given by the exponential map:

$$\mathfrak{g} = \{\mathfrak{h} \in M(n) : \ e^{t \cdot \mathfrak{h}} \in G \ \forall t \in \mathbb{R}\}, \tag{1}$$

where $M(n)$ is the set of all $n \times n$ matrices. Each such element $\mathfrak{h}$ from equation 1 is called an *infinitesimal generator* of the Lie group $G$.

We introduce **LieGG**, a method to compute **Lie G**roup **G**enerators. LieGG extracts the infinitesimal generators of the Lie algebra for symmetries that a neural network learned from the data. To find the Lie algebra basis, we use the *discriminator* of the dataset, i.e. a function $F : \mathbb{R}^n \to \mathbb{R}$ such that $F(\mathbf{x}) = 0$ if and only if $\mathbf{x}$ is an element of the dataset $\mathcal{D}$. The discriminator function is naturally modeled by a neural network fitting the data subject to a downstream task. With this, we use the following criterion for a symmetry group:

**Theorem 3.1** (Theorem 2.8 [29])**.** *Let $G$ be a connected Lie group of linear transformations acting on an $n$-th dimensional manifold $\mathcal{X}$. Let $F : \mathcal{X} \to \mathbb{R}^l$, $l \leq n$, define a system of algebraic equations:*

$$F_\nu(\mathbf{x}) = 0, \ \nu = 1, \cdots, l$$

*and assume that the system is of maximal rank, meaning that the Jacobian matrix $\left(\frac{\partial F_\nu}{\partial x_k}\right)$ is of rank $l$ for every solution $\mathbf{x}$ of the system. Then $G$ is a symmetry group of the system if and only if*

$$\sum_{i=1,j=1}^{i=n,j=n} \frac{\partial F_\nu}{\partial x_i} \cdot \mathfrak{h}_{ij} \cdot x_j = 0, \text{ whenever } F_\nu(\mathbf{x}) = 0,$$

*for $\nu = 1, \cdots, l$ and every infinitesimal generator $\mathfrak{h}$ of $G$, where $\mathfrak{h}_{ij}$ is an element of the matrix $\mathfrak{h}$ in the $i$-th row and $j$-th column.*

**Example**   We illustrate the practical significance of the theorem with an example. Suppose the dataset $D$ lies on a sphere in $\mathbb{R}^n$:

$$D = \{\mathbf{x} \in \mathbb{R}^n : x_1^2 + \cdots + x_n^2 = 1\}.$$

The discriminator for this example is the function $F(\mathbf{x}) = x_1^2 + \cdots + x_n^2 - 1$. According to the theorem, to find the Lie algebra of the symmetry group, we can find a basis of the solution of the following linear equations, where $a_{ij}$ are the variables of interest:

$$\sum_{i,j} x_i \cdot a_{ij} \cdot x_j = 0,$$

when $x_1^2 + \cdots + x_n^2 = 1$. We assert that the solutions are the family of matrices $A$ such that $A + A^T = 0$. Indeed, for $\mathbf{x} : x_i = 1, x_j = 0, j \neq i$, the condition means that $a_{ii} = 0$. For $\mathbf{x} : x_i = \frac{1}{\sqrt{2}}, x_j = \frac{1}{\sqrt{2}}, x_k = 0, k \neq i, j$, it follows that $a_{ij} + a_{ji} = 0$. On the other hand, $A = -A^T$ implies $\sum_{i \neq j} (x_i \cdot a_{ij} \cdot x_j + x_j \cdot a_{ji} \cdot x_i) = \sum_{i \neq j} (x_i \cdot a_{ij} \cdot x_j - x_i \cdot a_{ij} \cdot x_j) = 0$.

Note that the Lie algebra of the matrices $\{A : A + A^T = 0\}$ corresponds to the Lie group of matrices $\{B : B \cdot B^T = E\}$ which is the rotation group. Thus, the symmetry of a sphere is the rotation group.

## 4   Method

To compute a Lie algebra given a discriminator function $F : \mathbb{R}^n \to \mathbb{R}$ we use Theorem 3.1 and solve the system of linear equations, where each equation corresponds to one element in the dataset:

$$\sum_{i=1,j=1}^{i=n,j=n} \frac{\partial F}{\partial x_i} \cdot \mathfrak{h}_{ij} \cdot x_j = 0, \text{ for each point in the dataset.} \tag{2}$$

This is a system of linear equations with $n^2$ variables $\mathfrak{h}_{ij}$ and the number of equations equal to the number of points in the dataset. We can cast solving such system with respect to $\mathfrak{h}$ as a problem of finding a nullspace of the matrix $\mathcal{E}$. The matrix $\mathcal{E}$ has the number of rows equal to the number of points in the dataset and $n^2$ columns. Multiplying $\mathcal{E}$ with the vectorized representation of $\mathfrak{h}$ yields the system of equations in 2. We call the matrix $\mathcal{E}$ the *network polarization matrix*.

The problem of finding the nullspace basis of a matrix is a standard problem in the numerical analysis. It can be efficiently solved using the Singular Value Decomposition (SVD). Recall that we can write matrix $\mathcal{E}$ using the SVD in the following way $\mathcal{E} = U\Sigma V^T$, where $U, V$ - are orthogonal matrices and $\Sigma$ is a diagonal matrix with decreasing singular values. Thus, the columns in $V$ corresponding to nearly-zero singular values encode the nullspace of the system of equations in 2 and hence form a Lie algebra basis.

Thereby, in practice, the calculation of LieGG consists of 3 steps: (i) training a neural network, (ii) calculation of the polarization matrix $\mathcal{E}$, (iii) computing singular vectors corresponding to almost zero singular values of the polarization matrix $\mathcal{E}$.

### 4.1   Computing a Lie algebra of a group acting on $\mathbb{R}^2$

Symmetries play an important role in computer vision problems, and we will describe LieGG for this case in more details. In various computer vision problems, it is assumed that the symmetry group $G$ changes images from the dataset by acting on $\mathbb{R}^2$. For example, convolutional networks [24] and recently proposed equivariant networks for rotations [42, 39] assume that the group acts as a

translation or rotation of an image. These neural networks perceive each image as a function $f$ with non-zero values on some compact subset $\Omega \subset \mathbb{R}^2$, where the elements of the subset correspond to spatial coordinates. Thus, the space of images $\mathcal{I}$ is the space of all functions that map spatial coordinated to pixel values. In other words: $\mathcal{I} := \{f, \ f : \Omega \to \mathbb{R}\}$. A vision model $F : \mathcal{I} \to \mathbb{R}$ is a function on such a space of images.

We view an image is a function on a pixel grid $(\mathbf{x}_1, \cdots, \mathbf{x}_n)$ of the domain $\Omega$, i.e. the image can be identified with the point in $\mathbb{R}^n$, which encodes the pixel values. In this case the model takes $n$ real values as an input: $F(f(\mathbf{x}_1), \cdots, f(\mathbf{x}_n))$.

It is important to note that Theorem 3.1 assumes that the group $G$ acts continuously on the data manifold. However, this assumption is violated for the image data as an image is a non-continuous function of coordinates due to the discrete pixel grid. We overcome the discontinuity issue by preprocessing images with Gaussian smoothing.

To use LieGG for the image data, we explicitly write the condition that the matrix $\mathfrak{h}$ is an infinitesimal generator for the image data $\mathcal{I}$ on the grid $(\mathbf{x}_1, \cdots, \mathbf{x}_n)$. The condition reads as follows:

$$F(f(e^{\mathfrak{h} \cdot t} \circ \mathbf{x}_1), \cdots, f(e^{\mathfrak{h} \cdot t} \circ \mathbf{x}_n)) = 0 \leftrightarrow \sum_{p=1,k=1,j=1}^{p=n,k=2,j=2} \frac{\partial F}{\partial f(\mathbf{x}_p)} \cdot \frac{\partial f(\mathbf{x}_p)}{\partial x_p^k} \cdot \mathfrak{h}_{kj} \cdot x_p^j = 0 \quad (3)$$

where $t \in \mathbb{R}$ and $x_p^j$ corresponds to the projection of $\mathbf{x}_p$ onto the j-th coordinate axis. The resulting condition is the equation in 4 variables $\mathfrak{h}_{kj}, \ k = 1, 2; \ j = 1, 2$. Hence, the polarization matrix has 4 columns. Thus, in equation 3 a space of potential image symmetries is reduced to a 4-dimensional spatial subgroup. This is to provide a prior for LieGG to retrieve meaningful symmetries from for images as the full-scale space of potential symmetries is overwhelming for the image data. This is a strong yet proven to useful prior for vision models [6, 19, 42, 39, 20].

In practice, the calculation of LieGG for vision models consists of the following steps: (i) preprocessing images by doing Gaussian smoothing, (ii) training a neural network, (iii) computing the polarization matrix using equation 3, (iv) computing singular vectors corresponding to almost zero singular values of the polarization matrix.

## 4.2 Symmetry bias and symmetry variance

The scope of LieGG is not limited to retrieving the generator of a learned symmetry group. With the polarization matrix $\mathcal{E}$, we can also evaluate the degree of the neural network's invariance to a learned symmetry. Furthermore, if a true symmetry in the data is known, we can apply our method to analyze how close is a learned symmetry to the true one. The former criterion we alias *symmetry variance*, and the later - *symmetry bias*.

Firstly, we show how to estimate the degree of the network invariance to the element $g$ of a Lie group:

$$R(g) := \mathbb{E}(F(g \cdot \mathbf{x}) - F(\mathbf{x}))^2 \sim \frac{1}{|D|} \sum_{\mathbf{x}_i \in D} (F(g \cdot \mathbf{x}_i) - F(\mathbf{x}_i))^2 := \tilde{R}(g).$$

Let $\mathfrak{h}$ be an element from Lie algebra that corresponds to $g$, i.e.

$$\tilde{R}(g) = \frac{1}{|D|} \sum_{\mathbf{x}_i \in D} (F(\exp(\mathfrak{h}t) \cdot \mathbf{x}_i) - F(\mathbf{x}_i))^2 =$$

$$\frac{1}{|D|} \sum_{\mathbf{x}_i \in D} (t \cdot \sum_{k,j} \frac{\partial F(\mathbf{x}_i)}{\partial x_i^k} \cdot \mathfrak{h}_{kj} \cdot x_i^j + \mathcal{O}(t^2))^2 =$$

$$\frac{1}{|D|} \sum_{\mathbf{x}_i \in D} t^2 \cdot (\sum_{k,j} \frac{\partial F(\mathbf{x}_i)}{\partial x_i^k} \cdot \mathfrak{h}_{kj} \cdot x_i^j)^2 + \mathcal{O}(t^3)$$

The matrix with the $i$-th row equal to $\frac{\partial F(\mathbf{x}_i)}{\partial x_i^k} \cdot x_i^j$ equates to the polarization matrix $\mathcal{E}$. Then

$$\tilde{R}(g) = \frac{1}{|D|}(t^2 \cdot \left\|\mathcal{E}\bar{\mathfrak{h}}\right\|_2^2 + \mathcal{O}(t^3)),$$

where $\bar{\mathfrak{h}}$ is the vectorized representation of the element of the Lie algebra. To calculate the value of $\mathcal{E}\bar{\mathfrak{h}}$ we use the SVD of the matrix $\mathcal{E}$. Let $\mathcal{E} = U\Sigma V^T$, where $U, V$ - are semi-unitary matrices and $\Sigma$ is a diagonal matrix. With this, we can estimate $\tilde{R}(g)$ in the following way:

$$\tilde{R}(g) \sim \frac{1}{|D|}\left\|\mathcal{E}\bar{\mathfrak{h}}\right\|_2^2 = \frac{1}{|D|}\sum_l \sigma_l^2(V^T\bar{\mathfrak{h}})_l^2 \tag{4}$$

where $\sigma_l$ is the $l$-th singular value of the network polarization matrix $\mathcal{E}$.

In particular, the equation 4 reveals that singular values of the network polarization matrix reflect the degree of invariance to transformations encoded by the corresponding singular vectors. With this, we use the smallest singular value of the network polarization matrix to define the *symmetry variance* as $\mathcal{V}(\mathcal{E}) = \sigma_{min}(\mathcal{E})^2/|D|$. Symmetry variance estimates the degree of invariance in the direction of the symmetry, a network learned to be most invariant to. Moreover, when the true symmetries are known, the difference between the singular vectors and the generator of the true symmetry group equates to the *symmetry bias*, i.e. $\mathcal{B}_i(\mathcal{E}, \mathcal{G}) = ||\bar{v}_i(\mathcal{E}) - \mathcal{G}||_F$, where $\bar{v}_i(\mathcal{E})$ returns the i-th singular vector reshaped to a matrix, and $\mathcal{G}$ is a ground truth symmetry group generator closest to the $\bar{v}_i(\mathcal{E})$, e.g. the closest skew-symmetric matrix for the rotation group.

Note that the proposed symmetry variance allows measuring learned invariance without the knowledge of a symmetry generator beforehand. This is in contrast to Gruver *et al.* [16], where the knowledge of a symmetry generator is required to measure learned equivariance.

## 5 Experiments

### 5.1 Retrieving symmetries from the synthetic regression

Firstly, we conduct the experiment on the synthetic problem to test the ability of our method to accurately retrieve a symmetry learned by a network. To do so, we adopt the invariant regression task from [14] with clean $O(5)$ symmetry built-in. The regression function $f : \mathbb{R}^{2\times5} \to \mathbb{R}$ is given by:

$$f(\mathbf{x}_1, \mathbf{x}_2) = \sin(\|\mathbf{x}_1\|_2) - 0.5 \cdot \|\mathbf{x}_2\|_2^3 + \frac{\mathbf{x}_1^T\mathbf{x}_2}{\|\mathbf{x}_1\|_2\|\mathbf{x}_2\|_2} \tag{5}$$

With this, we construct the training dataset to fit the regression function with the multi-layer perceptron. Here the MLP is the unconstrained feed-forward neural network with no prior knowledge of symmetries in the data. Once the model is converged, we apply our method to analyze how close

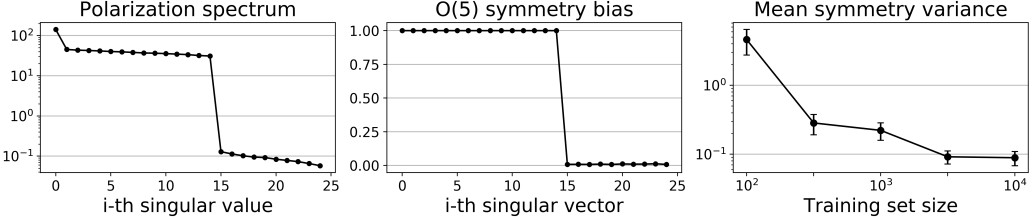

Figure 2: Singular values of the network polarization matrix and the symmetry biases per singular vector of the multi-layer perceptron fitted on $O(5)$ invariant regression task. **(right)** Comparing the mean symmetry variance for various training set sizes.

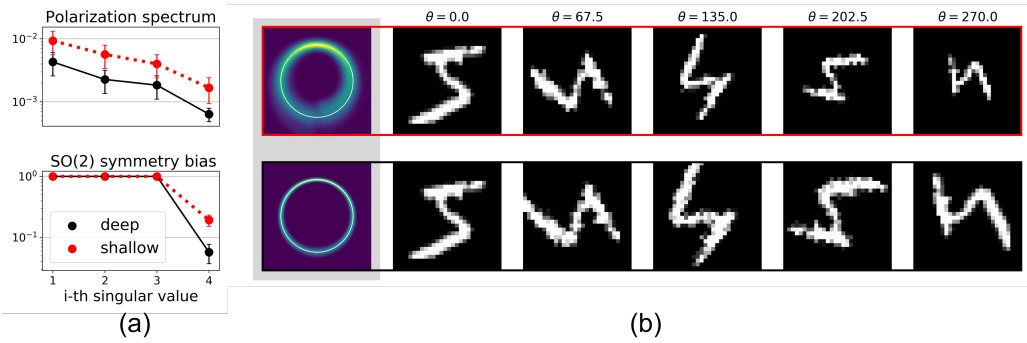

Figure 3: **(a)** Symmetrical properties of the shallow and deep networks: singular values of the network polarization matrix and SO(2) symmetry bias. **(b)** Shaded: the kernel density of learned 2D rotations on a perfect circle. Non shaded: visualizing learned Lie algebras for various rotation angles (in columns). Top row: the shallow network. Bottom row: the deeper network.

are retrieved symmetries to clean O(5) group (symmetry bias), and how invariant is the network to the learned symmetries (symmetry variance). We also study how the ability of the network to learn symmetries depends on the number of available training samples. We report the results in Figure 2.

As can be seen from Figure 2a, the spectrum of the network polarization matrix is zeroing, indicating that the model becomes invariant to the transformations encoded by the corresponding singular vectors. Also, these singular vectors are the generators of the rotation group as indicated by zero symmetry bias with respect to O(5) group (Figure 2b). We also observed that the network accurately learns the symmetry group generators even with a small training set size. However, a larger training set size leads to a smaller symmetry variance, i.e. a higher degree of invariance. For an additional validation, we used LieGG to extract symmetries from the randomly initialized O(5) invariant EMLP [14] model. For this case, LieGG recorded zero symmetry variance and zero symmetry bias with respect to O(5) group without any model pre-training.

The experiment demonstrates that, on the synthetic regression task, our method can accurately retrieve symmetries learned or built-in. When symmetries are learned, the larger training dataset facilitates learning invariances for the model.

## 5.2 Retrieving symmetries from the rotation MNIST

In this experiment, we test the ability of our method to retrieve learned symmetries on the rotation MNIST dataset [23]. We train the feed-forward network to output the class of a digit invariant to 2D in-plane rotations. The symmetry group that the model should learn is thus SO(2). We note that different from the synthetic regression task, the symmetries in the rotation MNIST are much noisier since the rotations are coarsened by the projection onto a pixel grid. In addition, the network has a classification error, hence it serves as only an approximate data discriminator function. In this scenario, we test how well our method can retrieve noisy SO(2) symmetry.

We experiment with two architectures: the shallow 2-layer perceptron and the deeper 6-layer network with 4 times number of parameters. Once the network's performance plateaus on the validation split, we terminate the training, and apply our method to retrieve learned symmetries. We plot the spectrum of the network polarization matrix, SO(2) symmetry bias and visualize learned symmetries for the shallow and deep models in Figure 3.

From Figure 3a we observe that for both shallow and deep models, our method retrieves the rotation symmetry as indicated by the zeroing symmetry bias with respect to SO(2) group. Also, we noticed that the symmetry learned by the shallow model is noisier than the symmetry learned by the deeper network as indicated by the higher symmetry bias. We visualize this in Figure 3b by sampling learned Lie algebras and applying them to transform an input image. We see that the symmetry learned by the shallow model diverges from a clean rotation and also contains scaling and shearing. We also see, that on the rotation-MNIST, the spectrum of the network polarization matrix is zeroing not as fast as

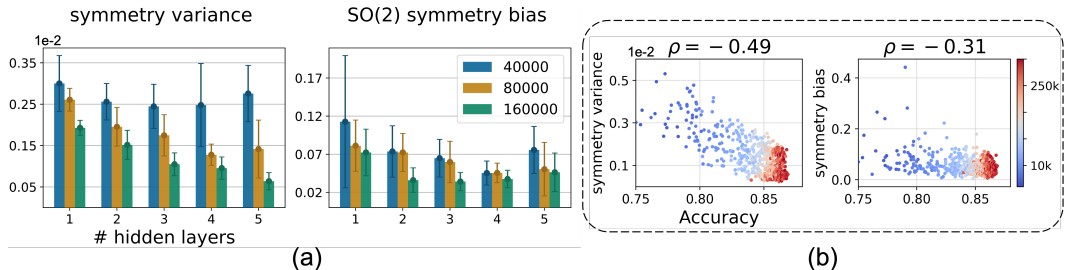

Figure 4: **(a)** Symmetry variance and SO(2) symmetry bias for the networks with different width, depth and number of parameters. **(b)** The correlation plot of the symmetry variance and the symmetry bias versus the test accuracy of the models. Each point corresponds to one network. The color indicates the number of parameters.

in the synthetic case. It indicates that the models, although do learn the rotation symmetry, do not become fully invariant to it.

This experiment demonstrates that our method can retrieve symmetries from the data even when an imprecise discriminator function is used.

### 5.3 Symmetries in networks with different configurations

Observing the differences in symmetrical properties for different network configurations in Experiment 5.2, we further conduct the study on how the symmetry variance and the symmetry bias depend on width, depth, and a number of parameters in a neural network. In this experiment, we are interested to know if for some network configurations it is easier to learn symmetries from the data.

To obtain different configurations, we sample a number of parameters from the range of $[40000, 200000]$ with the step size of $20000$, and vary the depth of a network from $1$ to $5$ hidden layers. We train all the networks until convergence on the validation split and apply LieGG to retrieve the symmetries and to record the symmetry variance and the symmetry bias. We also track the accuracy of the models to correlate it with the symmetry variance and the symmetry bias.

The results are reported in Figure 4. We summarize the analysis of how the configurations of the network influence its symmetrical properties:

**Number of parameters:** we observe that the networks with more parameters tend to learn to the higher degree of invariance than their shallow counterparts as indicated by consistently smaller symmetry variance. Also, we see that the networks with fewer parameters learn less precise symmetries as indicated by the higher symmetry bias.

**Depth / width:** our results indicate that sufficiently parameterized deep networks are more capable to learn invariances than the wide ones with the same number of parameters. In other words, deeper networks become more invariant in the direction of learned symmetries. At the same time, the same only partially holds for the quality of a learned symmetry group as indicated by the fluctuating symmetry bias, when the depth $> 3$. Overall, the network with a balanced width and depth achieves the smallest symmetry bias.

**Symmetry vs. accuracy:** we also test how the symmetry variance and the symmetry bias correlate with the performance of the models. As can be seen from Figure 4b, the symmetry variance and bias have moderate negative correlation coefficients $\rho_{\text{var}} = -0.49$ (*p-value* $< 1e^{-4}$) and $\rho_{\text{bias}} = -0.31$ (*p-value* $< 1e^{-4}$) with the classification accuracy respectively. The trend indicates that more accurate models tend to have lower symmetry variance and lower symmetry bias. However, the correlation is not strict. That is, it is possible to have neural networks, which learn symmetries differently, but perform equally well during the test.

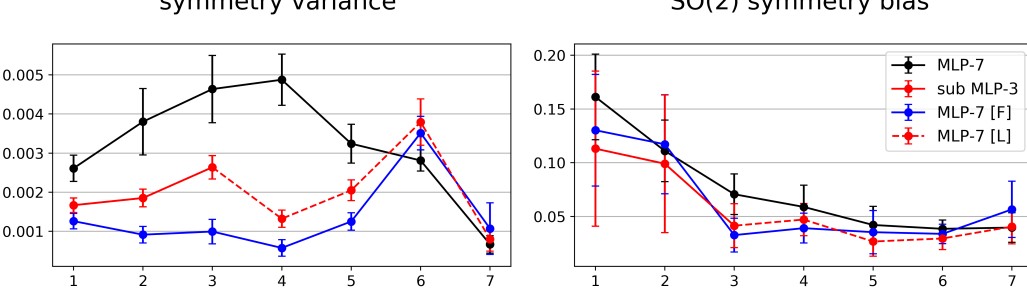

Figure 5: Symmetry variance and SO(2) symmetry bias measured layer-wise for the network trained on the rotation-MNIST dataset. Different colors correspond to different training regimes of the model. Training scenarios are: **MLP-7** for the 7-layer perceptron, **sub MLP-3** for the 3-layer sub-network, **MLP-7[F]** for the joint fine-tuning, **MLP-7[L]** for the layer-wise training.

## 5.4 Tuning layer-wise symmetries in deep networks

In the previous experiment, we analyzed how the symmetrical properties of the networks depend on the configuration of the model. Now, we ask if by modifying a training regime, we can facilitate learning symmetries for the network. In this experiment, we evaluate two known approaches to robust training: (i) pre-training with fine-tuning, and (ii) layer-wise training [5]. We also evaluate plain end-to-end training as a baseline.

To simulate the pre-training of a part of the network weights, we take randomly initialized 7 layers perceptron (mother network) and extract the sub-network that consists of the first 3 layers. We then attach the classification head to the sub-network and train it independently from the mother network on the rotation-MNIST dataset. After that, we re-inject the learned weights into the mother network and retrain the model. For the fine-tuning, we train all of the weights jointly. For the layer-wise training, we freeze the pre-trained part and only train the remaining layers. We then apply our method to study the symmetrical properties of the models trained with each of the regimes. We plot the symmetry variance and the symmetry bias after each layer of the network in Figure 5.

**Final and pen-ultimate layer symmetries:** from Figure 5 we observe that for the different training regimes, the network learns approximately the same symmetries at the same degree of invariance in the final layer. Surprisingly, we also noticed that in the pen-ultimate layer, the symmetries for all of the training regimes come very close as measured by the symmetry variance and bias. Since the last layers are directly responsible for providing outputs for the downstream task, we hypothesize that *there exist a preferred optimal symmetry level for a neural network that depends on an architecture and the data, but not from a training regime of a model*.

**Intermediate symmetries:** on the other hand, we observed that *symmetries in the intermediate layers of the network do depend on the training regime*. Firstly, the pre-trained sub-network learns more accurate symmetry with the higher level of invariance compared to when it is trained as a part of the deeper model. We attribute it to the fact that symmetries are defined on a level of training labels, hence invariances learned by the model in the final layer should be kept at a minimum [40]. Secondly, after the fine-tuning or the layer-wise training, symmetries in the intermediate layers of the network are both more precise and utilized at the higher degree of invariance as indicated by the smaller symmetry bias and variance respectively. It indicates that *we can facilitate learning intermediate invariances in a neural network* by using a pre-training followed by the fine-tuning or the layer-wise learning approaches. This can be useful for a transfer learning when it is important to transfer as much inductive bias as possible into other models or to other tasks.

# 6 Conclusion

**Summary** We present LieGG - the approach to retrieve symmetries learned by neural networks. Conveniently, our method extracts symmetries directly in the form of corresponding Lie group generators. This allows to explicitly obtain symmetries learned by a neural network without specifying any set of transformations of interest beforehand. We next introduce the symmetry variance and the symmetry bias to evaluate symmetrical properties of neural networks by analyzing the network polarization matrix computed by LieGG. We make use of our approach by studying how the quality of the symmetry learned by the neural network depends on a network's configuration and a training regime. By improving the interpretability of symmetry learning, we identify configurations and training regimes that facilitate training invariances in neural networks.

**Limitations and Future Work** The proposed method can facilitate the interpretability of neural networks and help to design efficient ways of incorporating symmetries into a model. However, some limitations remain. In particular, LieGG is currently limited to instances of $GL(n)$ connected group, which hinders the analysis of complex invariances, e.g. where a transformation can not be described as a group action. It is an intriguing direction for future research to extend a mechanism of retrieving symmetries to more complex transformations. Nonetheless, LieGG provides an important building block on the way to increasing the interpretability of symmetry learning in neural networks. Also, in our analysis, we focus on the specific family of models, i.e. feed-forward multi-layer perceptions trained on the datasets with well-controlled factors of variations. We assume that further large-scale analysis of LieGG in complex scenarios and with more advanced models can provide a more general understanding of symmetry learning in neural networks.

# 7 Acknowledgments and Funding Disclosure

We would like to thank Evgenii Egorov and Volker Fischer for their valuable comments and discussions. We thank the reviewers for their insightful feedback and helpful suggestions that improved our paper. This work has been financially supported by Bosch, the University of Amsterdam, and the allowance of Top consortia for Knowledge and Innovation (TKIs) from the Netherlands Ministry of Economic Affairs and Climate Policy.

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
