# A Supplementary material

## A.1 Derivation of Theorem 3.1

Let $G$ be a connected Lie group of transformations acting on the $n$-th dimensional manifold $\mathcal{X}$. Let $F : \mathcal{X} \to \mathbb{R}^l$, $l \leq n$. With this, we define a system of algebraic equations:

$$F_\nu(\mathbf{x}) = 0, \ \nu = 1, \cdots, l,$$

and assume that the system is of maximal rank, meaning that the Jacobian matrix $\left( \frac{\partial F_\nu}{\partial x_k} \right)$ is of rank $l$ for every solution $\mathbf{x}$ of the system. Then $G$ is a symmetry group of the system if and only if:

$$\mathfrak{h}[F_\nu(\mathbf{x})] = 0, \ \text{whenever} \ F_\nu(\mathbf{x}) = 0,$$

for every infinitesimal generator $\mathfrak{h}$ of $G$.

In our special case, the infinitesimal generator $\mathfrak{h}$ is a matrix, which generates the one parameter group of transformations equal to $\exp(\mathfrak{h} \cdot t)$. That means that in the theorem, $\mathfrak{h}$ is an element of the symmetry group if and only if $\frac{dF_\nu(\exp(\mathfrak{h} \cdot t) \cdot \mathbf{x})}{dt}|_{t=0} = 0$, whenever $F_\nu(\mathbf{x}) = 0$. But

$$\frac{dF_\nu(\exp(\mathfrak{h} \cdot t) \cdot \mathbf{x})}{dt}\Big|_{t=0} = \sum_{i=1}^{i=n} \frac{\partial F_\nu(\mathbf{x})}{\partial x_i} \cdot \frac{(\exp(\mathfrak{h} \cdot t) \cdot \mathbf{x})_i}{dt}\Big|_{t=0} =$$

$$\sum_{i=1}^{n} \frac{\partial F_\nu(\mathbf{x})}{\partial x_i} \cdot \sum_{j=1}^{n} \mathfrak{h}_{ij} \cdot x_j = \sum_{i=1,j=1}^{i=n,j=n} \frac{\partial F_\nu}{\partial x_i} \cdot \mathfrak{h}_{ij} \cdot x_j$$

## A.2 LieGG sample complexity

LieGG computation requires finding a nullspace of the network polarization matrix. The dimensionality of the network polarization matrix depends on the number of samples in a dataset. Since the dimensionality of a full-scale dataset may be prohibitively large, we conduct the sample complexity study to investigate if the usage of all samples in a dataset is necessary to effectively retrieve the symmetries learned by a neural network.

We conduct the empirical sample complexity study for O(5) synthetic regression and rotation-MNIST classification tasks. For the synthetic regression, we take the trained model from A.3.1 and compute the symmetry variance and bias using only a fraction of the training samples. We then compare the results with the symmetry variance and bias computed from all available training data. The resulting empirical sample complexity is presented in Figure 1. For the rotation-MNIST, we follow the same procedure and study the sample complexity of LieGG applied to retrieve symmetries from the trained 6-layer perceptron from A.3.2. The resulting empirical sample complexity for the rotation-MNIST is presented in Figure 2. Note that in both cases, the model is trained using all available training data. The sample complexity study is only to investigate how many samples are needed to effectively extract the symmetries from the trained model.

For the synthetic regression task, it suffices to use only $0.25\%$ of the training samples to extract the symmetry as precisely as when utilizing all available training data. For the more challenging rotation-MNIST, we have to use at least $5\%$ of the training samples to obtain a precise estimation of the symmetry bias. The results suggest that we can effectively extract the learned symmetry using only a fraction of the data to create the network polarization matrix.

## A.3 Experiments

Here we provide additional experiments and details on the experiments in the main paper. All of the experiments are performed with Pytorch `1.7.0` [3] on Nvidia Titan X GPU with the CUDA version `10.1`. The results and error bars are reported over 8 fixed random seeds `[1-8]`.

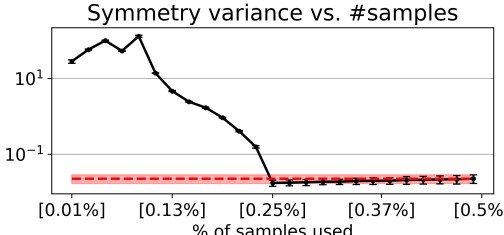 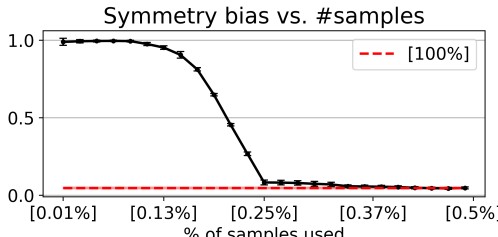

Figure 1: The sample complexity of LieGG for the $O(5)$ synthetic regression task. The red line indicates the symmetry variance and the symmetry bias computed using $100\%$ of available samples.

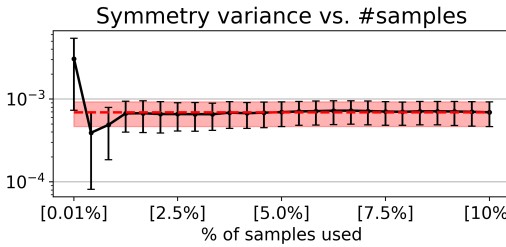 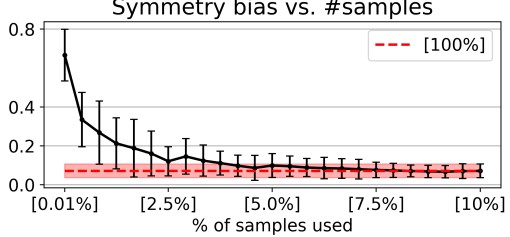

Figure 2: The sample complexity of LieGG for the rotation-MNIST classification task. The red line indicates the symmetry variance and the symmetry bias computed using $100\%$ of available samples.

### A.3.1 Retrieving symmetries from the synthetic regression

To fit the regression function we employ the multi-layer perceptron that consists of linear layers followed by Swish [4] activations with the following dimensions: $(10 \times 32) \rightarrow 3 * (32 \times 32) \rightarrow (32 \times 1)$. The network takes input coordinates and outputs the predicted function value. We use a mean squared error loss to supervise the model. We utilize Adam optimizer [2] with a learning rate of $10^{-3}$ and train the network for a total of $min(900000/N, 1000)$ epochs, where $N$ is the size of the training dataset. [1] reports this training schedule is enough for convergence.

To compute the mean symmetry variance per a training set size, we average the last 10 singular values of the network polarization matrix. These singular values correspond to the $O(5)$ symmetry as indicated by the corresponding symmetry biases.

### A.3.2 Retrieving symmetries from the rotation-MNIST

We employ two models: shallow (2-layer) and deep (6-layer) perceptrons with $40000$ and $160000$ parameters respectively. The hidden dimensions for the shallow and deep networks are $47$ and $116$ respectively. Both models are trained for $300$ epochs with Adam optimizer with the learning rate set to $10^{-3}$. We use a cross-entropy loss to supervise the model.

Symmetries are extracted from the models with the highest validation accuracy over $300$ training epochs. Additionally, to account for a possible difference in the magnitudes of the network outputs, we normalize the network output logits to be unit L2 norm.

### A.3.3 Retrieving symmetries from the rotation-augmented CIFAR10

We follow the Experiment $5.2$ of the main paper for the rotation-augmented CIFAR10 dataset to check if LieGG retrieves symmetries from the dataset, which is more complex than the rotation-MNIST.

We employ the 7-layer perceptron with a hidden dimension equal to $400$. We train the model for $300$ epochs on the CIFAR10 dataset augmented with random rotations. We use Adam optimizer with the learning rate of $10^{-3}$. We select the best epoch model on the validation subset and visualize the

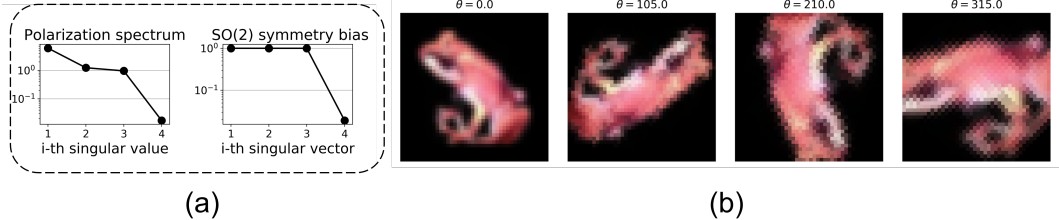

(a)                                                       (b)

Figure 3: **(a)** The spectrum of the network polarization matrix and the symmetry bias for the MLP trained on the CIFAR10. **(b)** the visualization of learned Lie algebras for various rotation angles.

symmetrical properties retrieved in Figure 3. Similar to the rotation-MNIST, LieGG retrieves the correct symmetry as indicated by zeroing symmetry bias.

### A.3.4   Symmetries in networks with different configurations

To evaluate networks with various configurations (width, depth, number of parameters), we consider the following family of architectures: $(784 \times hdim) \rightarrow p * (hdim \times hdim) \rightarrow (hdim \times 10)$, where $p$ stands for the number of hidden layers and $hdim$ is the dimension of a hidden layer. Given the required number of parameters and the number of hidden layers, we can calculate the hidden dimension of the network. By varying the number of parameters and the number of hidden layers we create wide and deep networks.

We follow the same procedure as in A.3.2 to train the models and extract symmetries.

### A.4   LieGG implementation

We provide the PyTorch implementation of the network polarization matrix computation used in the synthetic regression and the rotation-MNIST experiments. Symmetries are retrieved by performing a singular-value decomposition of the network polarization matrix as described in Section 4 of the main paper.

▶ **synthetic regression:**

```python
def polarization_matrix(model, data, dim = 5):
    # data: torch.FloatTensor(B, 2*dim)
    B = data.shape[0]
    data.requires_grad = True
    data.retain_grad()

    # compute network grads
    model.eval()
    y_pred = model(data)
    y_pred.backward(gradient=torch.ones_like(y_pred))

    # get grads and data per input dimension
    dF_1 = data.grad[...,:dim].view(B, dim, 1)
    data_1 = data[...,:dim].view(B, 1, dim)

    dF_2 = data.grad[...,dim:].view(B, dim, 1)
    data_2 = data[...,dim:].view(B, 1, dim)

    # collect into the network polarization matrix
    C = torch.bmm(dF_1, data_1) + torch.bmm(dF_2, data_2)

    return C
```

▶ **rotation-MNIST:**

```python
def polarization_matrix_R2(model, data):
    # LieGG implementation with the groups acting on R^2
    # data: torch.FloatTensor(B, 28, 28)

    B, H, W = data.shape

    # compute image grads
    data_grad_x = data[:, 1:, :-1] - data[:, :-1, :-1]
    data_grad_y = data[:, :-1, 1:] - data[:, :-1, :-1]
    dI = torch.stack([data_grad_x, data_grad_y], -1)

    # compute network grads
    data = data.reshape(B, -1)
    data.requires_grad = True
    data.retain_grad()

    output = model(data)
    output.backward(torch.ones_like(output))

    dF = data.grad.reshape(B, H, W)
    dF = dF[:, :-1, :-1]

    # coordinate mask
    xy = torch.meshgrid(torch.arange(0, H-1), torch.arange(0, H-1))
    xy = torch.stack(xy, -1)
    xy = xy / (H // 2) - 1

    # collect into the network polarization matrix
    C = dF[..., None, None] * dI[..., None] * xy[None, :, :, None, :]
    C = C.view(B, -1, 2, 2).sum(1)

    return C
```