# OpenReview forum: "LieGG: Studying Learned Lie Group Generators"
_NeurIPS.cc/2022/Conference — NeurIPS 2022 Accept_

### Official Review · Reviewer_Pv3i · 2022-07-03

**Rating:** 6
**Confidence:** 4
**Soundness:** 3 good
**Presentation:** 3 good
**Contribution:** 3 good

**Summary:**

The paper presents a method, LieGG, for identifying learned in variances in a trained model.  Specifically, by differentiating the model with respect to its inputs and looking at these derivatives at various inputs points, a polarization matrix is constructed which has as its nullspace, generators of a Lie algebra of invariances. This can be learned using SVD.  The authors apply their method to MLPs trained on a sythentic function with O(5) symmetry and Rotated MNIST which has SO(2) symmetry.  In both cases, the learned model displays the expected invariance.  The authors also analyze the effect of network size, shape, and trained on learned symmetry.

**Questions:**


## Minor Points

- Line 93: Technically, it is not enough to know the Lie algebra to know the symmetry group.  For example, we discard any discrete symmetries.
- Line 114: 2 is missing from the equation
- Line 126: call it _the_ network ...
- Equation 3 after Line 150: There is no sum over images only over pixel locations.  Should there be a sum over images f as well?
- Equations after Line 163: How is t sampled?
- Equations after Line 163: In the second line of the equation, I think it should be t^2
- Equations after Line 163: In the third line of the equation, I think there is a missing parenthesis.
- Equation after Line 164: Should it be O(t^4)?
- Line 165: Why is this only an estimate?
- Equation after Line 167: Should h_l be squared?
- Line 170: "Last" singular values is imprecise.
- Line 182: with a _closest_
- Line 188: the input space is $\mathbb{R}^{10}$ or $\mathbb{R}^{2 \times 5}$. It doesn't really make sense to use both multiplicative and exponential notation for the same thing.
- Figure 5: describe abbreviations in caption

## Post-Response Edit

I appreciate the authors response to my questions and maintain my positive score.

**Limitations:**

The limitations section is good, but the authors could be clearer about what "more complex transformations" means.  The assumptions in Theorem 3 and resulting limitations should also be clarified.

**Strengths And Weaknesses:**

## Strength

- This work is very timely and the question is important.  It is becoming increasingly clear that symmetry is a key part of deep learning and the idea that symmetry can be learned directly from data has motivated several recent works on symmetry discovery.  The hope is that neural networks may be useful in e.g. scientific domains for discovering interpretable findings about different systems from data.
- The method is an effective application of the theory of symmetries of differential equations.  The background is clearly presented and theorem 2.8 form [37] is nicely explained with an example.
- The method is able to explicitly find symmetries of learned models.  These symmetries are in the form of Lie algebra generators.  This is similar to the form of the symmetries found by (Dehmamy, 2021), however, an advantage of the current work is that no special architecture is necessary in order to learn the symmetry.
- It could be said the proposed method is not really for demonstrating that neural networks learn symmetry (see first weakness/question), but for learning invariances of the ground truth function from data.  At first, I wondered if these invariances could be estimated directly from data without training a neural network first.  However, it seems that training a neural network gives an effective way to estimate df/dx_i in equation 2 which would otherwise be difficult to do directly from data alone.  In this light, I think the proposed method is quite an effective way to discover invariances in data indirectly.
- The finding (Line 244-245) that deeper networks learn symmetry better as the capacity is increased versus shallow networks is interesting and seems useful in network design.


## Weakness / Questions
- My main question about the method is that it's unclear what it means for a model to learn symmetry independently of learning the task function accurately.  Consider a regression problem.  If the model F trains well to approximate a ground truth function which is invariant with RMSE error at most \epsilon, then it seems to me the symmetry variance (invariance error) is bounded by 2\epsilon.  That is, the model has symmetry because it learned the ground truth and that has symmetry.  In this case, I would not really say the model did any particular invariance learning.  If it could be demonstrated that the model had even lower symmetry variance that expected given the accuracy or was capable of extrapolation based on the invariance, then I would grant the model had learned symmetry.  This question is somewhat examined in "Symmetry vs. accuracy" (Line 250), but here the conclusion seems to support my suspicion that any learned symmetry is really an emergent consequence of accurately fitting the ground truth function.
- I don't understand the second sentence of the abstract.  If a network first learns symmetry wouldn't it be starting to learn from no symmetry?  I presume the point of this sentence is perhaps to contrast with methods such as equivariant neural networks in which symmetry is encoded into the network as a prior, but I don't think it is very clear.
- Theorem 3.1, Line 107-108. I looked at the reference [37] from which this theorem is quoted.  This statement appears to be a special case of the generality stated there, but with some missing assumptions.  The hypothesis here says n-dimensional manifold X, however, the equation given does not refer to local coordinates and so I believe assumes X=R^k and G \subset GL_k(R) and acts by matrix multiplication on X.  If I understand correctly, this puts in the setting of [37, Example 1.28b] with a linear vector field.  That is, we are assuming symmetries are global linear symmetries of the space.  This is definitely still a useful level of generality to consider, but it does exclude some cases.  For example, F(x,y) = y - x^2.  I'd recommend clarifying the assumptions and deriving the special case from the general theorem in the appendix.
- Section 4.1 has some unclear writing.  The idea of image as function (Line 140-141) can be clearer.  Line 141:  A function should be defined either in terms of spaces, $X \to Y$ or elements $x \mapsto y$ or both.  This mixes both.  The notation in the equation between Line 146 and Line 147 is unclear.  Is the goal actually to describe a group action on the space of images?  In that case, I would define $(g \cdot f)(x) = f(g^{-1} \cdot x)$.  In particular, I would not use the same notation for the the group G and the action map $a \colon G \times X \to X$.  Lastly, what is really going on here (it seems to me) is that the space of potential symmetries on the full space of images is too high-dimensional for the current method to be effective and so you have limited yourself to the 4-dimensional subgroup of spatial symmetries.  This is a strong prior over the potential invariances you are looking for.
- Line 161:  It would be very helpful to include a formal definition of symmetry bias and symmetry variance.
- Line 224-225: I'm not sure I agree with the conclusion about the smoothness of the spectrum of the polarization matrix or with the conclusion that the method can robustly find noisy symmetries.  In this case there is a very strong prior of the symmetry group being within a 4-dimensional subgroup of spatial symmetries. It's hard to evaluate smoothness over only 4 data points and the strength of this inductive bias could be the reason the model performs well here.
- "Intermediate Symmetries" (Line 276): In an equivariant neural network, only the last layer or last layers are usually used to project to an invariant representation.  Intermediate representations usually display equivariance instead of invariance with respect to some latent group representation type.  In that sense, it is perhaps not surprising that intermediate layers do not show invariance.

---

> ### Author Response · Authors · 2022-08-02
> **Reply to Review Pv3i.**
>
> - Learning symmetry & task function
>
> In this work, we differentiate between symmetry variance and symmetry bias. In the reviewer's example, the symmetry variance is indeed bounded by $2\epsilon$. However, the value of symmetry bias is unbounded. Consider the case when a network trained on a limited dataset memorizes the training subset. In this case, the learned invariance may differ from the assumed invariance, and symmetry bias will have a high value. If the dataset contains enough transformed samples to reconstruct the whole orbit of the group, we can expect a low enough value of symmetry bias as well.
>
> - Theorem 3.1 and assumptions
>
> We agree with the reviewer. Indeed, we consider the special case of the theorem, when the group $G$ acts linearly. Specifically, we assume that the manifold $X$ is embedded in a high dimensional space $\mathbb{R}^m$, such that each point $\textbf{x}\in X$ can be described using coordinates in $\mathbb{R}^m$. We also assume that the action of the group $G$ is defined $\textit{only}$ on $X$. In this case, as the reviewer correctly notices, the group action is linear, hence it is determined by a linear vector field, but $\textit{only}$ on $X$ (i.e. on the zero space of the algebraic equations given in the theorem). Thus, we do not assume global linear symmetries of the space. We add the derivation of the special case of the theorem into the supplementary material.
>
> - Notation
>
> We improve the notation in Section 4.1 to include the reviewer’s suggestions.
>
> - The subgroup of spatial symmetries
>
> Indeed, the full-scale space of the potential symmetries is overwhelming for image data. To cope with that, we use the 4-dimensional subgroup of spatial symmetries, which is a standard subgroup for 2D vision tasks (e.g. [1,2,3,4] consider the same subgroup). This is indeed a strong yet proven to be useful prior in vision. We add clarification about the prior in Section 4.1.
>
> - Formal definition for symmetry variance/bias
>
> We add the formal definitions of the symmetry variance and bias in Section 4.2 in the manuscript
>
> - Eq.(3) sum over images
>
> Eq.(3) shows an equation that should hold for any image. Thus, there should not be a sum over the images, but rather it is a system of equations with one equation per image.
>
> - Other
>
> We add the clarifications to make the wording more precise, and we fix minor points in the manuscript.

---

### Official Review · Reviewer_BrAS · 2022-07-11

**Rating:** 6
**Confidence:** 3
**Soundness:** 2 fair
**Presentation:** 2 fair
**Contribution:** 2 fair

**Summary:**

The authors propose LieGG, a method for recovering the Lie group generators corresponding to the symmetries of a trained network. Using their approach, the authors can measure how close a recovered generator is to the true symmetry and to what degree a network is invariant to that transformation. The authors evaluate on synthetic data and MNIST using MLP networks of different depth with known symmetries in the data.

**Questions:**

* What is v_p in equation 3?
* How does LieGG perform without pre-processing with Gaussian smoothing? Is spatial smoothness of the input data a requirement for LieGG to perform well? This ablation would be nice to have.
* What is the p-value corresponding to the correlations in Figure 4b? Please report these in the manuscript.
* How does this approach scale with dataset size (computational complexity of SVD + sample complexity of LieGG)? It would be nice to include experiments that measure these properties.
* Is it possible use LieGG to co-train with network weights to decrease the symmetry variance of a network? For example, (Connor and Rozell 2020) show that after a fine-tuning step with learned Lie group generators, the corresponding Lie group provides a significantly better model of MNIST digit rotation.

**Limitations:**

* The authors metric for symmetry variance seems to fall off in performance as the depth of a network is increased. Perhaps this can be attributed to the approximation to the matrix exponential employed on line 163. The weakness of this approximation has been noted in the literature before in (Rao et al. 1999) vs (Culpepper and Olshausen 2009).

**Strengths And Weaknesses:**

Strengths
* The authors propose a theoretically motivated algorithm for recovering a Lie group generator from a deep neural network that is novel to the best of my knowledge. They evaluate the efficacy of their algorithm on MNIST rotation to show that a deep network trained on rotated MNIST digits implicitly becomes invariant to rotation.
* The authors propose valuable metrics to evaluate the quality of a learned Lie group and to measure how invariant a model is to a given Lie group.

Weaknesses
* The authors argue that networks trained with “hard [coded] symmetry” may “not always lead to a better generalization” on lines 20-21, but they never support this claim with experimental data. It would be nice for the authors to compare using their proposed metrics between deep networks that implicitly learn symmetry and those that explicitly incorporate symmetry, such as (Finzi et al. 2021).
* The experimental results on the invariance of MLPs at different depths are of moderate significance. I would have hoped the authors test their method with larger models and datasets with more advanced invariances. For example, it would be interesting to apply an augmentation to samples from a dataset like CIFAR10 and use LieGG to measure whether the deep network becomes invariant to that augmentation.

Misc fixes:
* There are several spelling and grammatical mistakes in the experiments section. Please re-read and revise. Here are a few:
    * “We the MLP” line 178
    * “closes” line 182
    * “How the ability of the network to learn” line 193
    * “The symmetry learn by the” line 220
* The authors should consider references to a part of the literature on learning Lie group generators from data & incorporating them into networks to encourage invariance (a different, yet related task to recovering Lie group generators from a network to evaluate invariance), such as ("Learning Lie Groups for Invariant Visual Perception,” Rao et al. 1999), (“Learning transport operators for image manifold,” Culpepper and Olshausen 2009), (“An Unsupervised Algorithm For Learning Lie Groups,” Sohl-Dickstein et al. 2010), and (“Representing Closed Transformation Paths in Encoded Network Latent Space,” Connor and Rozell 2020).
* Please explicitly state which training scenario corresponds with each entry in the legend of Figure 5.

---

> ### Author Response · Authors · 2022-08-02
> **Reply to Review BrAS**
>
> - Hard-coded symmetries
>
> We draw inspiration from [9,10,11,12] where authors experimentally demonstrated that the soft symmetries often deliver better test accuracy than the hard-coded ones.
>
> - Extracting explicitly incorporated symmetries
>
> We agree that applying the LieGG for the networks with explicitly incorporated symmetries is a good validation of the method. We employ the LieGG to extract symmetries from the randomly initialized $\text{O}(5)$ invariant EMLP (Finzi et al.). For the EMLP, the LieGG recorded zero symmetry variance and zero symmetry bias with respect to the $\text{O}(5)$ group without any model pre-training. We add the result into Experiment 5.2.
>
> - Advanced models and invariances
>
> We purposely conducted experiments with feed-forward models on the small dataset with well-controllable factors of variation to isolate the effect of learning invariances in neural networks. We agree that further analysis of the LieGG in complex scenarios can provide a more general understanding of the invariance learning in deep networks. Due to time limitations, we have to leave a large-scale experimental study of the LieGG for future work. We clarify the Limitation and Future work section to inspire further research with the LieGG.
>
> - Related work
>
> Thank you for the recommended papers. We added them to the revised version of the manuscript. We suppose It will give a reader a better understanding of the current state of the field and how Lie Algebra-Lie Group correspondence was and is used in ML.
>
> - Gaussian smoothing
>
> Gaussian smoothing is required when the LieGG is applied to discretized image data. This is needed to make sure that the action from group G is continuous with respect to image coordinates as required by Theorem 3.1. We add the clarification to the manuscript to additionally emphasize the role of Gaussian smoothing for discretized image data.
>
> - Clarifications
>
> We adjust the notation in Eq.(3): $v_p \rightarrow \textbf{x}_p [j]$, where $\textbf{x}_p [j]$ corresponds to the projection of $\textbf{x}_p$ onto the j-th coordinate axis. We report the p-values for the correlations in Figure 4b. We add Figure 5 legend description.
>
>
> - Co-training the LieGG with network weights
>
> All operations used to compute the symmetry variance are differentiable. Thus, we suppose that the LieGG can be used as an auxiliary objective to facilitate invariance learning in deep networks.

---

> > ### Author Response · Authors · 2022-08-09
> > **Sample complexity**
> >
> > Before the closure of the rebuttal/discussion period, we want to address the sample complexity question. At this time, we have finished the sample complexity study of the proposed method.
> >
> > The results suggest that we can effectively extract the learned symmetry using only a fraction of the dataset. For the synthetic regression task, it suffices to use only $0.25$% of the training samples to extract the symmetry as precisely as when utilizing all available training data. For a more challenging rotation-MNIST, we have to use at least $5$% of the training samples to obtain a precise estimation of the symmetry bias. The sample complexity study is added into the supplementary material.
> >
> > $\text{We thank the reviewers for their time and insightful comments.}$

---

### Official Review · Reviewer_erES · 2022-07-11

**Rating:** 7
**Confidence:** 3
**Soundness:** 3 good
**Presentation:** 3 good
**Contribution:** 3 good

**Summary:**

This paper proposes learned Lie Group Generators (LieGGs) to investigate symmetries encoded in neural networks inversely. It is based on the Lie group-Lie algebra correspondence, which describes an arbitrary matrix Lie group as an exponential map of the corresponding Lie algebra. The authors exploit the fact that infinitesimal generators of a Lie algebra $\mathfrak{g}$ of a matrix Lie group $G$ acts on data $D$ should satisfy a certain algebraic equation (Theorem 3.1) when there is a $G$-invariant mapping function $F: X \to \mathbb{R}$ that satisfies $F(x) = 0$ where $x \in D$. Because a neural network is an approximation of such $F$ (if the underlying data is $G$-invariant and the neural network learns the data well), the learned symmetries encoded in neural networks can be inversely retrieved by using the proposed method. The authors validate their proposed approach for a simple synthetic problem and rotated MNIST. Also, the authors formulate two metrics that can measure a degree of encoded symmetries, i.e., symmetry bias and variance, based on their approach.

**Questions:**

As I mentioned, I am generally satisfied with the paper. Some minor comments are the followings:

1. Can the authors elaborate on the effect of the Gaussian smoothing, e.g., is it critical when using the binary rotated MNIST directly?
2. In (2), isn't $i = j=n$?
3. In (3), it seems that the underlying group structure is assumed to be a one-parameter sub-group $\exp(t \mathfrak{h})$ parameterized by $t \in \mathbb{R}$ which is not fully general. Please elaborate on whether the proposed method is applicable for a general $GL(n)$ or its one-parameter subgroup only.
4. Assume a subset of the rotated MNIST consists of '6's and '9's. '6's and '9's yield identical $G$-orbits while they are not G-invariant with respect to the labeling distribution. Can the proposed method work well for such a subset?
4. Can the proposed method be used for investigating $G$-equivariance?

**Limitations:**

The authors state the limitation (limited to $GL(n)$ group) of the proposed method in the section Conclusion. I think that this is an acceptable limitation considering $GL(n)$ is a major and interesting group structure for various problems.

**Strengths And Weaknesses:**

[Strengths]

Extracting as well as learning symmetries is a definitely crucial problem in the modern deep learning field. While most of the previous works focus on the latter problem, this paper addresses the former one based on a theoretically convincing method, namely Lie algebra. The paper is well-written and the presentation is easy-to-follow. The proposed symmetry bias and variance might be potentially useful metrics to measure the degree of symmetries when building a new $G$-invariant model.

[Weaknesses]

I did not find any critical drawback in this paper. Possible weaknesses include:

There is no comparison with competitors. Especially, a comparison with the following paper might be important:

N. Dehmamy, R. Walters, Y. Liu, D. Wang, and R. Yu. Automatic Symmetry Discovery with Lie Algebra Convolutional Network. NeurIPS 2021.

Dehmamy et al. also use the concept of Lie algebra to deal with arbitrary group symmetries, and as its title indicates, it seems their method might be used for the recovery of underlying symmetries encoded (if I am wrong please correct me). I think the lack of the comparison is not a critical issue because the proposed method is more general, i.e., can be applied to arbitrary neural network architectures - but at least a simple comparison with Lie algebra CNN for the rotation MNIST seems to need for the completeness of the work.

Rotated MNIST is only a dataset benchmarked in this paper. Because both MNIST and $SO(2)$ are under relatively simple settings, I am not fully convinced whether the proposed method works well for other datasets or group structures. Possible benchmark candidates might include more complicated images with $SE(2)$.

---

> ### Author Response · Authors · 2022-08-02
> **Reply to Review erES**
>
> - Comparison with L-conv
>
> The reviewer is right, the method of N. Dehmamy et al. allows to retrieve transformations learned by the network from the training data. However, in this case, a symmetry analysis is possible for networks with L-conv layers, which is not the case for the models we considered in our paper.
>
> - Other datasets
>
> We agree with the reviewer that evaluation with more datasets would be beneficial for the paper. We train the MLP on the CIFAR10 dataset with a rotation augmentation. We then use the LieGG to extract learned symmetries as in Experiment 5.3. Our results suggest that the LieGG can successfully extract the symmetries learned by the network on rotation-augmented CIFAR10. We add this experiment into the supplementary material.
>
> - “6’s” and “9’s” orbits
>
> You are right that “6”s and “9”s will have (almost) identical orbits. It was demonstrated in [3] that rotation equivariant and invariant models perform way better than standard counterparts on rotated-MNIST, and their 98%+ accuracy indicates that models with the symmetry built-in distinguish between “6”s and “9”s. What we can tell from the experiments in [3] is that the best performing models contain symmetries to additional groups apart from SO(2) which generate different orbits for these numbers. In this case, an additional symmetry group generator is learned.
>
> - Gaussian smoothing
>
> Since images are initially discretized, Gaussian smoothing is a requirement for the LieGG to make sure the action from group G is continuous with respect to image coordinates as required by Theorem 3.1. We clarify the text in Section 4.1 to additionally emphasize the role of Gaussian smoothing in case of a discretized image data.
>
> - Eq. (2) summation indices
>
> The reviewer is right, $i$ and $j$ vary from $1$ to $n$.
>
> - One parameter subgroup $\text{exp}(t \mathfrak{h})$
>
> Every element of the Lie algebra generates a one-parameter subgroup that satisfies Eq.(3). Thus, Eq.(3) shows the condition that $\mathfrak{h}$ belongs to the Lie algebra. The set of all such infinitesimal generators generates a Lie algebra, which may consist of several linear independent vectors.
>
>
> - G-equivariance
>
> The theoretical grounds of the LieGG can be extended to investigate G-equivariance.

---

> > ### Comment · Reviewer_erES · 2022-08-09
> > **Reply to the authors**
> >
> > I appreciate the authors’ response to my questions. It answered some of my concerns.
> >
> > I still think the paper is worthy of publication, thus keep my score.

---

### Meta-Review · Area_Chair_g5yg · 2022-08-27

**Recommendation:** Accept
**Confidence:** Certain

**Metareview:**

The paper proposes a novel technique to extract symmetry inductive biases from data that can be applied to any neural network architecture. Please include more discussions with the related work and possible experimental comparisons in the updated version.

**Award:**

No

---

### Decision · Program_Chairs · 2022-09-14

Accept